# Bispecific Antibody Format and the Organization of Immunological Synapses in T Cell-Redirecting Strategies for Cancer Immunotherapy

**DOI:** 10.3390/pharmaceutics15010132

**Published:** 2022-12-30

**Authors:** Carlos Carrasco-Padilla, Alicia Hernaiz-Esteban, Luis Álvarez-Vallina, Oscar Aguilar-Sopeña, Pedro Roda-Navarro

**Affiliations:** 1Department of Immunology, Ophthalmology and ENT, School of Medicine, Universidad Complutense de Madrid and 12 de Octubre Health Research Institute (imas12), 28040 Madrid, Spain; 2Cancer Immunotherapy Unit (UNICA), Department of Immunology, Hospital Universitario 12 de Octubre, 28009 Madrid, Spain; 3Immuno-Oncology and Immunotherapy Group, 12 de Octubre Health Research Institute (imas12), 28009 Madrid, Spain; 4H12O-CNIO Cancer Immunotherapy Clinical Research Unit, Spanish National Cancer Research Centre (CNIO), 28029 Madrid, Spain

**Keywords:** IS, T cell, immunotherapy, bsAb, CAR, TCR

## Abstract

T cell-redirecting strategies have emerged as effective cancer immunotherapy approaches. Bispecific antibodies (bsAbs) are designed to specifically recruit T cells to the tumor microenvironment and induce the assembly of the immunological synapse (IS) between T cells and cancer cells or antigen-presenting cells. The way that the quality of the IS might predict the effectiveness of T cell-redirecting strategies, including those mediated by bsAbs or by chimeric antigen receptors (CAR)-T cells, is currently under discussion. Here we review the organization of the canonical IS assembled during natural antigenic stimulation through the T cell receptor (TCR) and to what extent different bsAbs induce T cell activation, canonical IS organization, and effector function. Then, we discuss how the biochemical parameters of different formats of bsAbs affect the effectivity of generating an antigen-induced canonical IS. Finally, the quality of the IS assembled by bsAbs and monoclonal antibodies or CAR-T cells are compared, and strategies to improve bsAb-mediated T cell-redirecting strategies are discussed.

## 1. Introduction

Bispecific antibodies (bsAbs) are molecules with binding sites for two different antigens or two different epitopes on the same antigen [1]. They represent valuable immunotherapy tools and can be designed to redirect T cells to cancer cells, although some of them have been also used to treat other diseases, such as hemophilia A [2] or Alzheimer’s disease [3]. The first application of bsAbs in cancer immunotherapy was for redirecting T cells toward tumor cells. T cell-engaging bsAbs (TCEs) are specifically engineered to simultaneously bind to a predefined tumor-associated antigen (TAA) on the surface of cancer cells and to one of the extracellular CD3 subunits (usually CD3ε) of the T cell receptor (TCR) expressed on the surface of T cells, leading to the release of preformed cytotoxic proteins, such as perforin and granzymes, as well as cytokines [4,5]. TCEs mediate a major histocompatibility complex (MHC)-independent T cell activation and are applicable to all patients regardless of their MHC haplotype. Other bsAbs are designed to target co-stimulatory molecules [6] co-inhibitory checkpoints [7,8] enhance T cell activation, or target TAAs to block dual signaling pathways necessary for tumor growth [9,10]. Regarding the mechanism of action of TCEs, establishing an artificial link between the TCR and a user-defined TAA facilitates both the recruitment of T cells to the tumor microenvironment (TME) and the establishment of the immunological synapse (IS) [11].

In hematological tumors, the administration of bsAbs has been a major step forward in clinical practice [12]. Blinatumomab, a CD19×CD3 TCE designed for the treatment of relapsed or refractory B cell acute lymphoblastic leukemia (B-ALL), is the most remarkable example [13,14,15]. Solid tumors present additional challenges to bsAb-based therapies, given that the TME is strongly immunosuppressive [16], and the majority of known TAAs are also expressed at low levels on normal tissues [17], leading to severe on-target off-tumor toxicities [18]. A further shortcoming of Fc-free bsAb-based therapies is the need for continuous infusion due to rapid clearance from the circulation [19]. Table 1 lists the bsAbs approved by the Food and Drug Administration (FDA) and/or the European Medicines Administration (EMA), or in clinical trials for the treatment of hematological or solid tumors.

This review initially discusses the importance of the IS in achieving physiological immune responses and then focuses on the ability of different bsAbs to trigger T cell activation and induce IS assembly and effector functions. Secondly, it analyzes how the biochemical characteristics of the different bsAb formats influence the organization of the IS and T cell responses. Then, it compares the results obtained by bsAbs with those generated by CAR-T cells or monoclonal antibodies (mAbs). Finally, different strategies that are expected to benefit bsAb-mediated cancer immunotherapy are discussed.

## 2. Immune Synapse Formation and Pathology

The IS is a specialized adhesion formed between T cells and antigen-presenting cells (APCs) that is essential to sustain T cell activation and effector function [32]. Seminal studies showed that cognate interactions conducted to stable mature immune synapses composed of concentric Supramolecular Activation Clusters (SMACs) (Figure 1A). A central (c)SMAC was shown to contain the TCR and signaling molecules, such as PKCθ [33], while a peripheral (p)SMAC contained the integrin LFA-1 and components of the cytoskeleton important for the stability of the adhesion [34,35]. More recent data show that the early TCR signaling is triggered in peripheral microclusters (MCs), which move towards the cSMAC, where signaling is terminated [36,37,38]. This centripetal movement is mediated by a retrograde flow of filamentous actin (F-actin), organized at the so-called distal (d)SMAC and by the contractility of actomyosin arcs organized at the pSMAC [39,40]. The contraction of actomyosin arcs also guides integrin clusters to their correct position at the pSMAC [39]. F-actin retrograde flow is also critical for sustaining PLCγ1 activation during early TCR activation signaling [41]. In order to maintain the TCR downstream signaling, the phosphatase CD45 is excluded from the TCR activation sites at MCs [37].

Currently, it is well known that the cSMAC contains a region for costimulation-generated PKC signals [42] and a region where the TCR is endocytosed [36]. To accomplish optimal sustained activating signaling, the delivery of the TCR from the endosomal compartment to replace endocytosed and degraded molecules during activation is required [43]. The cSMAC is also the site for the polarization of the microtubule organizing center (MTOC). This mediates sustained signaling [44] and the polarization of the endosomal compartment for the secretion of cytokines [45], lytic granules [46], and exosomes [47]. From the cSMAC, ectosomes are also released, which bud from the plasma membrane of the T cell [48]. It has been proposed that different kinds of extracellular vesicles and ectosomes released at the IS might assist the APC function [47,49,50].

Several of the described events in in vitro studies, most remarkably the formation of SMACs, have been robustly corroborated in vivo during immune responses against viruses and cancer [51,52,53]. Thus, it is expected that the organization of SMACs found in vitro might predict the role that this organization has during in vivo immune responses.

Due to the fact that the IS formation is responsible for T cell activation and effector function, the molecular dynamics at the IS are expected to have a significant impact on the quality of T cell responses. Consistent with this idea, patients diagnosed with multiple sclerosis and type 1 diabetes have self-reactive CD4 T cells, which have been shown to exhibit highly phosphorylated TCR MCs with reduced accumulation at the cSMAC of TCR-pMHC (peptide bound to the MHC) complexes [54]. Therefore, it seems that proper molecular dynamics at the IS are needed to achieve healthy T cell responses. It is then plausible to think that canonical IS organized in response to T cell-redirecting strategies would be essential to obtain effective and safe anti-tumor responses [55].

## 3. T Cell Is Organization and Effector Function Induced by bsAb

Seminal studies showed that a TCE effectively induced the assembly of an IS with a similar topology to the IS generated by antigen-stimulated T cells [11] (Figure 1). Offner and collaborators employed laser scanning confocal microscopy (LSCM) to assess the IS organization induced with a TCE specific for CD3 and the epithelial cellular adhesion molecule (EpCAM), which is overexpressed in epithelial cancer [56]. Analysis of the IS established between cytotoxic T cells and EpCAM-expressing cells showed that they were highly similar to those generated under antigen-specific stimulation. The formation of SMACs and the organization of perforin, LCK, PKCθ, CD3, LFA-1, and CD45 were nearly identical [11]. An EpCAMxCD3 TCE has shown therapeutic efficacy in malignant ascites in ovarian cancer patients, using intraperitoneal administration regimens [57].

The IS has been also studied in T cells redirected by a TCE targeting carcinoembryonic antigen (CEA), which is overexpressed in a variety of solid tumors [58]. In this case, the MTOC polarization and perforin accumulation was observed at the IS, which coexisted with tumor cell killing, secretion of cytokines and cytotoxic granules, and T cell proliferation in mice [23].

Another study demonstrated that a TCE specific for FcRH5, a B cell lineage-specific surface marker in multiple myeloma [59], induced the assembly of a canonical IS [60]. The analysis by LSCM of conjugates formed by FcRH5 expressing target cells and Jurkat (JK) CD4 T cells revealed the polarization of the FcRH5 antigen and ZAP70 to the IS and the efficient exclusion of CD45. Interestingly, the assembly of the IS correlated with an efficient triggering of in vitro early TCR signaling and cytotoxic function [60]. We also have previously shown efficient activating signaling in addition to the polarization of F-actin and CD3ε to the IS induced by an anti-epidermal growth factor receptor (EGFR) and anti-CD19 TCE. The organized IS showed a similar topology to antigen-stimulated IS and correlated with efficient T cell activation in vitro and in vivo [61,62,63]. Therefore, TCE-mediated T cell redirection seems to induce the assembly of a canonical IS, as well as effective T cell activating signaling and function similar to events induced during antigen stimulation of T cells.

Although TCEs have been proven to be useful, the engagement of CD137 (also known as tumor necrosis factor receptor superfamily member 9, TNFRSF9) on T cells (instead of CD3) and CD40 on APCs (instead of a TAA) has emerged as a new efficient immunotherapy [64,65]. Results obtained with the CD40xCD137 bsAb duobody have clearly shown this end [6]. This duobody was shown to render efficient polarization of LFA-1 to primary T cell/APC interactions, higher activation, and T cell activity in comparison with monovalent controlxCD40 and controlxCD137 bsAbs. It is expected that in this strategy, the engagement of CD137 would raise the activation of T cell clones specific for tumor antigens presented by APCs.

Thus, the use of bsAbs, which target CD3, or co-stimulatory molecules results in useful approaches to activate T cell responses against cancer. It should be noted that due to the expression of CD3^+^ in CD4^+^ and CD8^+^ T cells, the action of TCEs in the tumor site would not be restricted to CD8^+^ T cells, and the expected activation of CD4^+^ T cells might also contribute to the outcome of the immune response against the tumor. The action of TCEs in different CD3^+^ T cell subtypes should be investigated.

## 4. Format of bsAb and Immunological Synapse Organization

The clinical efficacy of bsAbs highly depends on the pharmacokinetics (PK), which is influenced by multiple parameters such as molecular size and avidity [66]. Intermediate-sized multivalent bsAbs aim to achieve a compromise between size and avidity in order to reach optimal PK and tumor targeting [16]. In this context, the use of heavy-chain-only immunoglobulins (VHHs) [67] is instrumental to increase the valence and functional affinity while preserving an adequate size for good tumor penetration [16]. Due to the nascent hypothesis that the success of T cell-redirecting strategies might depend on the quality of the IS [55] the extent to which the format of bsAbs influences the organization of the IS should be also taken into consideration. We discuss below how different biochemical parameters influence the assembly of the IS and the efficiency of T cell activation.

### 4.1. TAA-Binding

In 2017, an Fc-free TCE called ATTACK (asymmetric tandem trimerbody for T cell activation and cancer killing) was designed with trivalent and monovalent binding to EGFR and CD3ε, respectively (3 + 1 stoichiometry). Jurkat cells and EGFR-expressing cells were co-cultured and the IS assembly and T cell activation were compared with a TCE with a 1 + 1 stoichiometry with monovalent binding to EGFR and CD3ε, termed LiTE (light T-cell engager) [68]. LSCM showed a similar polarization of CD3ε, which was surrounded by a peripheral network of F-actin at the IS established under stimulation with both formats. Interestingly, however, the TAA avidity effect in the ATTACK format improved the efficiency of the polarized activating signaling to the IS, T cell activation, and T cell cytotoxic activity [62]. It is interesting to note that the optimal ATTACK size was not compromised by the increment of binding sites thanks to the use VHHs for the EGFR targeting. Therefore, the avidity of the bsAb might not only assist an adequate PK but also improves IS assembly and T cell activation.

Knowing that the avidity effect enhances activation, another interesting question was to determine whether binding to two different TAAs might assist T cell activation. The assembly of the IS by a trispecific T cell engager (TriTE) against EGFR, EpCAM, and CD3ε in comparison with two bispecific EGFRxCD3ε and EpCAMxCD3ε LiTEs has been also evaluated under the microscope by co-culturing Jurkat cells with colorectal cancer cells expressing EGFR and EpCAM. The number of cell interactions determined by F-actin polarization was significantly higher with the TriTE format, which also showed a higher T cell activation [69]. Therefore, interaction with two different TAAs also seems to assist T cell activation and IS assembly, with dual TAA-targeting also being an important feature preventing tumor escape by antigen loss caused by selective pressures from conventional single TAA-targeting TCEs.

### 4.2. bsAbs Size and Spatial Requirements

It has been suggested that large molecules cannot access the IS cleft. Different studies have shown that dextran particles and smaller antibodies enter the IS better than larger molecules and that a full-length mAb directed against a molecule secreted inside the IS cannot bind to its target [70,71]. Therefore, these studies suggest that the size of a bsAb intended to target IS-distributed molecules would be important to ensure the accessibility to the binding site. Thus, improving the PK of bsAbs while preserving their ability to enter activation sites at the IS would be a major challenge in this field. However, large bsAbs have been shown to properly trigger the IS. Cremasco et al. report the mechanism of action of a TCE with bivalent binding to CD20 and monovalent binding to CD3ε (2 + 1 stoichiometry), bearing an engineered Fc domain, which retains FcRn-driven half-life extension while reducing adverse immune induction by removal of FcɣR binding by site-specific mutations [29,72]. This TCE (glofitamab) with a molecular weight of 194 kDa is significantly larger than other bsAbs also designed for treating hematological cancers, such as blinatumomab (55 kDa). Despite being a large molecule, in vitro confocal imaging showed conjugate formation between CD8^+^ T cells and two CD20-expressing cell lines. Interestingly, glofitamab promoted a proper LFA-1 distribution to the established activating interactions [29]. These data highlight how large bsAbs with long half-lives are also able to induce IS formation and efficient target cell killing. These data suggest that, in addition to the size, other features of the bsAbs might also influence the ability to assemble the IS. It is likely that those TAAs, which naturally enter the IS might offer better options to accomplish this task.

Regarding the targeted TAA, a recent study has shown that the distance of the TAA epitope to the cell membrane is critical to achieving proper T cell activation, IS assembly, and cytotoxic activity against myeloma cells. The authors used several FcRH5-specific TCEs-recognizing epitopes located at different distances from the plasma membrane, demonstrating that stronger T cell activation and IS assembly resulted from proximal epitope targeting [60]. Therefore, it seems that not only the avidity or the number of targeted TAA should be taken into account but also the spatial location of the recognized epitope. Furthermore, probably not all the TAAs are equally suitable for triggering the assembly of a high-quality IS.

## 5. bsAbs versus mAbs

To achieve an effective and controlled T cell activation, the role of co-stimulatory and co-inhibitory immune checkpoint molecules is necessary. Immunotherapies based on checkpoint-blocking mAbs have achieved outstanding results in oncology [73,74]. Due to the fact that bsAbs can potentially improve the PK of mAbs [5], the comparison of the IS assembled by these two types of antibody molecules is important. The impact of combining two specificities for a TAA and an immune checkpoint in a single molecule has been studied by Gu and coworkers [75]. The authors showed that a PD-1xHER-2 bsAb can assemble the T cell IS compared to monospecific anti-HER-2 and anti-PD-1 mAbs, which did not induce any activating cell contact with tumor cells. The observed recruitment of PD-1 to the IS indicates a potential role in these strategies. Therefore, bsAbs combining PD-1- and TAA recognition could be the basis for new therapeutic agents.

## 6. bsAbs versus CARs

Chimeric antigen receptors (CARs) represent another T cell-redirecting strategy, in which a TAA-specific single chain scFv fused to a transmembrane and CD247 domains is transduced in T cells, making them effective killers for the TAA-expressing tumors [76] that can migrate to the TME to exert their cytotoxic function [77]. Consecutive improved generations of CAR-T cells with costimulatory and cytokine inducer domains have been developed over the past few years [78]. Although treatment with CAR-T cells has produced remarkable clinical responses in some hematological tumors, they are not absent of adverse effects [79]. We expect that a more natural IS organization might assist in achieving the aim of generating safer therapies. Thus, in order to achieve more effective and secure CAR-T cell-based therapies, an important question to be investigated might be the molecular dynamics mediating the CAR-T cell activation and how the IS is organized when interacting with tumor cells. Different studies have shown that during the assembly of the IS by CAR-T cells there is an efficient MTOC polarization and lytic granule delivery [80]. However, the actin cytoskeleton is not properly cleared from the cSMAC and signaling microclusters as well as LFA-1 displayed a disordered distribution. These properties correlate with more transient early signaling and unstable interactions that have posed the hypothesis of CAR-T cells being good serial killers [81].

When comparing CAR-T cell- and TCE-based therapies, the main advantages that CAR-T cells offer are the active cell trafficking to tumor sites and the costimulation-induced signaling, which are not feasible properties with current TCEs [82] (pros and cons of both approaches are indicated in Table 2). However, in contrast to TCEs, which are expected to engage any T cell at the TME, CAR-T cells cannot induce the TAA-specific activation of bystander tumor-infiltrating lymphocytes [82,83]. This seems to be a clear advantage of TCEs over CAR-based strategies and, given that the efficacy and safety of T cell-redirecting strategies might depend on the quality of the T cell activation and IS organization, the comparison of these bsAbs- and CAR-induced processes is an important question to be considered in future studies.

To conduct this comparative study properly, it is essential to use the same antibody clone, rendering the TCE and CAR functionally equivalent with the same specificity. We have recently performed this comparison with CD19-specific molecules, a well-characterized target for B cell neoplasias [84]. The 3D LSCM revealed that the IS organized in response to the CD19-specific TCE was more similar to a TCR-induced canonical IS than the IS assembled by the functional equivalent CD19-specific CAR-T. The TCE led to a similar CD3ε and F-actin organization at the IS compared to that observed in TCR-stimulated Jukat cells while actin clearance and CD3ε distribution in CAR-T Jurkat cells were not properly organized [63]. Importantly, similar results were also observed when primary cells were used [61]. Therefore, this analysis indicates a higher quality of the IS assembled by TCEs in comparison to CAR-T cells. However, further research will be needed to assess how different formats of TCEs and CAR-T cells might affect the mechanisms and kinetics of killing and the organization and functional contribution of other molecular components (such as integrins) of antigenic T cell activation.

## 7. Improving Therapies Based on bsAbs

### 7.1. Direct Secretion of bsAbs to the Tumor Site

Due to the extraordinary potency of T cell-mediated responses and the absence of tumor-specific antigens for the majority of cancers, there are significant risks associated with T cell responses against non-malignant tissues and/or systemic cytokine release-associated toxicities that are major barriers to the clinical application of bsAbs. To overcome these shortcomings, a novel T-cell redirection strategy using engineered T cells secreting T-cell redirection bsAb (STAb-T immunotherapy) has been devised. This approach might solve the rapid renal clearance associated with small-sized Fc-free bsAbs, enable a specific delivery of the bsAb to the TME, and produce better-tolerated antibodies due to the glycosylation pattern [85].

We have recently defined the topology of the IS induced by CD19xCD3 secreting STAb-T cells, compared to CD19-CAR T cells or the exogenous addition in the culture medium of purified CD19xCD3 blinatumomab. The IS assembly and early signaling displayed by the STAb Jurkat cells were similar to those assembled by TCR-stimulated Jurkat cells or purified blinatumomab [63]. In contrast, CAR-T cell signaling kinetics was shown to be more transient and the induced IS was disorganized as explained before, even when primary cells were used [61]. Results both in primary T cells and in vivo assays showed how canonical IS formation could be a predictor of treatment success, since the STAb-T cells induced more potent cytotoxicity, prevented tumor escape in vitro, and avoided leukemia relapse in vivo. The downmodulation of CD19 has been shown to be a mechanism of tumor escape from anti-CD19 therapies [86,87]. Interestingly, STAb-T cells did not cause the CD19 downmodulation found when CD19-CAR cells were used [61]. Then, STAb-T therapy might improve the PK of bsAbs, mediate the organization of a high-quality IS, trigger proper early signaling, and leave unaltered the expression of CD19 on tumor cells. Therefore, it is expected that the STAb-T strategy may have potential among next-generation cancer immunotherapy strategies.

### 7.2. Combining Different bsAbs

A combination of bsAbs is expected to improve immunotherapy by adding stimulatory effects that individual molecules cannot achieve. This could be useful because most TAA-specific TCEs lack domain binding to co-stimulatory molecules. Kantarjian H. and co-workers reported that the treatment of B cell precursor acute lymphoblastic leukemia (B-ALL) with blinatumomab induced a longer survival compared to chemotherapy [14], but the lack of co-stimulation leads to poor persistence of T cell responses and low rates of tumor-free patients after six months of treatment. Interestingly, the combination of a TAA-specific TCE providing signal 1 with a bsAb that mimics signal 2 by binding to a TAA on tumor cells and to a CD28 receptor on T cells, has been shown to achieve an enhanced IS formation and anti-tumor activity compared to the single use of the TAA-specific TCE [88]. This example shows how combining different bsAbs may help to overcome the problem associated with the lack of co-stimulation during bsAb-mediated cancer immunotherapy.

## 8. Conclusions and Final Remarks

Both hematological and solid tumors have been shown to be potentially treatable with bsAbs (Table 1). It is becoming increasingly clear that high-quality IS formation is a good predictor of successful cancer cell killing [23,60,61,62,63]. Therefore, a detailed analysis of the molecular reactions organized at the artificial IS will help to predict the efficiency and safety of therapies based on novel bsAb or CAR formats [89]. To achieve this goal with high spatial and temporal definition, we should take advantage of high-resolution microscopy techniques along with the use of bsAb-coupled fluorescent proteins, which will also aid in the observation of the bsAb delivery to the IS and fate in live cells [90]. Therefore, bsAbs should be designed with the appropriate format, assuring a compromise between size, avidity, targeted TAA, and epitope location, to achieve optimal PK and a high-quality IS.

In addition to the format, the direct targeting of cell-secreted bsAbs to tumor sites, as in the STAb-T approach, might solve problems associated with the short half-life of bsAb or its systemic toxicity. The bsAbs secreted by STAb-T cells can not only induce the assembly of the IS in the secreting cells but also in bystander T cells. Together, these features might generate an optimal scenario for an efficient and secure therapy.

Finally, the combination of different TAA-specific bsAbs, targeting CD3 and a co-stimulatory molecule, should certainly improve the efficacy of immunotherapies. Research on multispecific molecules targeting a TAA, CD3, and co-stimulatory or co-inhibitory checkpoint receptors is an active field of research, and it will be also necessary to understand the molecular dynamics mediating T cell activation in those approaches. We anticipate that preclinical in vitro studies focused on IS quality will assist the design of efficient and safe next-generation T cell redirecting strategies.

## Figures and Tables

**Figure 1 pharmaceutics-15-00132-f001:**
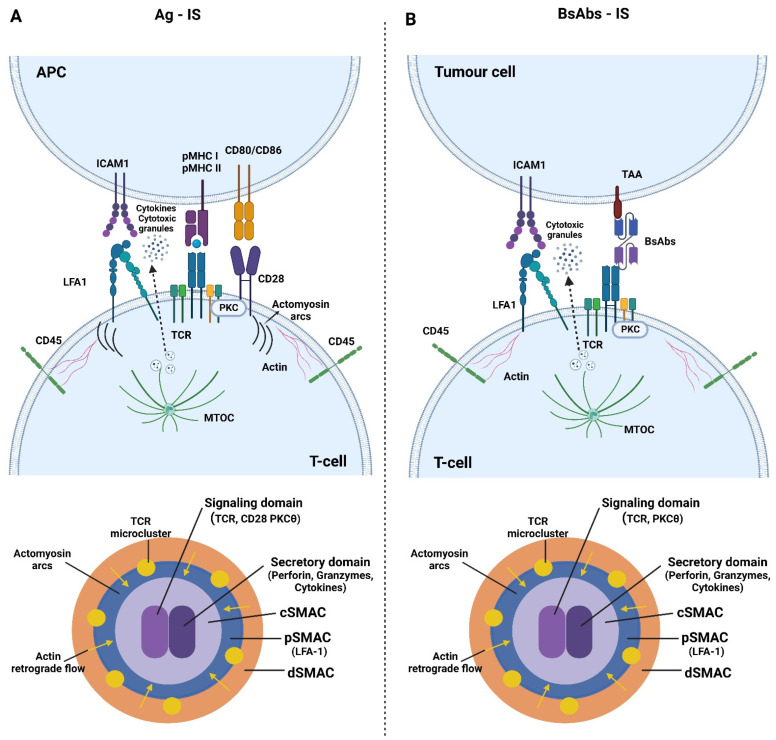
**Immunological synapse assembled in response to antigen or bsAbs.** (**A**) Schematic of the canonical IS assembled between a T cell and an APC presenting an antigenic peptide bound to the major histocompatibility complex (pMHC). pMHC I and II are indicated. However, for clarity, it is only represented the MHC class I. The main activating molecules, cytoskeleton components, and secretory granules or cytokines delivered at the contact interface are depicted in the upper panel. The principal components of the cSMAC, pSMAC, and dSMAC are indicated in the lower schematic. The signaling domain located in the cSMAC contains TCR microclusters (MCs), co-stimulation molecules such as CD28 and PKCθ. The secretory domain of the cSMAC secretes perforin- and granzyme-containing granules in the IS assembled by cytotoxic T cells and different types of cytokines in the IS assembled by helper T cells. (**B**) Schematic of the topology of the IS assembly induced by bsAbs between a T cell and a tumor cell. BsAbs typically engage the T cell by the CD3ε chain of the TCR and the tumor cell through a TAA. This leads to the establishment of antigen-stimulated canonical IS. Upper panel shows the main elements of the IS induced by bsAbs. The cSMAC, pSMAC, and dSMAC compositions are represented in the lower schematic. The signaling domain located in the cSMAC contains TCR MCs and signaling proteins such as PKCθ. Secretory domain of the cSMAC secretes lytic granules containing perforin and granzyme B.

**Table 1 pharmaceutics-15-00132-t001:** BsAbs approved by EMA/FDA and some of the bsAbs in clinical trials for the treatment of solid and hematological tumors. Formats, specificity of Target 1 and Target 2 binding (protein and gene name included), mechanism of action, type of cancer, and clinical phase are indicated. Abbreviations: EpCAM (epithelial cell adhesion molecule), GP100 (glycoprotein 100), EGFR (epidermal growth factor receptor), c-Met (tyrosine-protein kinase Met), HER3 (member 3 of EGFR family), HER2 (member 2 of EGFR family), LAG3 (Lymphocyte-activation gene 3), BCMA (B cell maturation antigen), DART (dual-affinity retargeting), ImmTACs (immune-mobilizing monoclonal TCRs against cancer), KIH (Knobs into holes), CRIB (Charge Repulsion Induced Bispecific), BiTE (bispecific T cell engager) and TriKe (tri-specific killer engagers). Data from the phase of clinical trials are derived from ClinicalTrials.gov and biochempeg.com (accessed on 4 November 2022) and identifiers are shown at the bottom of the table.

BsAbs Name	Format	Target Protein 1 (Gene; Cell)	Target Protein 2 (Gene; Cell)	Mechanism of Action	Type of Cancer		Phase (Identifier)
Catumaxomab	Triomab	CD3 (*CD3E*; T cell)	EpCAM (*EPCAM*; cancer cell)	Recruitment and activation of T cells [20]	Malignant ascites	Solid tumors	Approved by EMA *
Tebentafusp	ImmTAC		PMEL peptide 280–288 (*PMEL*; cancer cell)	Recruitment and activation of T cells [21,22]	Unresectable or metastatic uveal melanoma		Approved by EMA and FDA
RO6958688	CrossMab/KIH(IgG-like bsAbs)		CEA (*CEACAM5*; cancer cell)	Recruitment and activation of T cells [23]	CEA-positive tumors		Phase I (NCT02324257)
Amivantamab	Duobody	EGFR (*EGFR*; cancer cell)	METcMet (*MET*; cancer cell)	Blocking of dual signal pathways [9]	Non-small cell lung cancer (NSCLC)		Approved by EMA and FDA
SI-B001	IgG-(scFv)2		HER3 (*ERBB3*; cancer cell)	Blocking of dual signal pathways [10,24]	NSCLC		Phase I (NCT04603287)
GEN1402	Duobody	CD137 (*TNFRSF9*; T cell)	CD40 (*CD40*; APC)	Costimulating molecule engaging for efficient T cell activating signals [6]	NSCLC, Colorectal Cancer and Melanoma		Phase II (NCT04083599)
Zanidatamab	Asymmetric	HER2 (*ERBB2*; cancer cell)	HER2 (*ERBB2*; cancer cell)	Blocking of dual signal pathways [25]	Gastro-oesophagealadenocarcinoma		Phase II (NCT04513665)
Erfonrilimab	CRIB	CTLA-4 (*CTLA4*; T cell)	PD-L1 (*CD274*; cancer cell)	Blocking of immune checkpoints [7]	NSCLC and pancreatic ductal adenocarcinoma		Phase II (NCT03838848)
Tebotelimab	DART	PD1 (*CD80*; T cell)	LAG3 (*LAG3*; cancer cell)	Blocking of immune checkpoints [8]	Gastric Cancer		Phase III (NCT04082364)
Blinatumomab	BiTE	CD3 (*CD3E*; T cell)	CD19 (*CD19*; cancer cell)	Recruitment and activation of T cells [26]	Acute lymphoblastic leukaemia B	Haematological tumors	Approved by EMA and FDA
Mosunetuzumab	KIH (IgG1-like bsAb)		CD20 (*CD20*; cancer cell)	Recruitment and activation of T cells [27]	Relapsed or refractory follicular lymphoma		Approved by EMA
Glofitamab	CrossMab/KIH(IgG-like bsAbs)			Recruitment and activation of T cells [1,28,29]	Diffuse large B-cell lymphoma		Phase II/III (NCT03075696, NCT04408638)
Teclistamab	Duobody		BCMA (*TNFRSF17*; cancer cell)	Recruitment and activation of T cells [30]	Multiple myeloma		Approved by EMA
Flotetuzumab	DART		CD123 (*IL3RA*; cancer cell)	Recruitment and activation of T cells [31]	Acute myeloid leukaemia		Phase II (NCT03739606) *

* Withdrawn.

**Table 2 pharmaceutics-15-00132-t002:** PROS (green) and CONS (red) of TCE- and CAR-T cell-based therapies.

TCEs	CAR-T Cells
Absence of costimulatory signal	Costimulatory domain can be added in the CAR design
Passive and inefficient trafficking to tumor sites	Active and efficient trafficking to tumor sites
Polyclonal activation of bystander tumor-infiltrating lymphocytes (TILs)	Exclusive activation of CAR-expressing T cells, which cannot induce activation of bystander TILs
Short serum half-life and requirement of continuous intravenous administration	Long-term persistence
Cytokine release-associated toxicities	Cytokine release-associated toxicities
IS topology similar to canonical antigen-induced synapse	Non-canonical IS

## Data Availability

Not applicable.

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
