# Peer review of "Bispecific Antibody Format and the Organization of Immunological Synapses in T Cell-Redirecting Strategies for Cancer Immunotherapy"

_pharmaceutics, 2022, doi:10.3390/pharmaceutics15010132_

Round 1
Reviewer 1 Report
Please see the comments in the attached file

Reviewer 2 Report
The review is well documented and informative.
Comments
As the genes of the targets are known, it is recommended to use for each target the HGNC gene name (with the previous name as alias between parentheses the first time it is met in the manuscript).
Please indicate CD3E (instaead of CD3) whenever it is known.
Lines 35-36 : to one of the extracellular CD3 subunits (usually CD3e) of the T cell receptor (TCR) > to the extracellular domain of one of the CD3 coreceptors (usually CD3 epsilon (CD3E)) of the T cell receptor (TcR) ((TcR for TR + CD3))
Lines 38-39 : a major histocompatibility complex (MHC)-independent > a major histocompatibility (MH)-independent ((MHC for locus and haplotype, MH1 (MH class I) and MH2 (MH class II) for gene and protein))
Lines 21, 137, 262 : TCR >TcR (TcR for TR + CD3)
Line 34 : T cell engaging bsAb (TCE) > T cell engaging (TCE) bsAb
Line 39 : a major histocompatibility complex (MHC)-independent > a major histocompatibility (MH)-independent
Line 74 F-actin > filamentous actin (F-actin)
Line 95 : TCR-pMHC (peptide bound to the MHC) complexes > TR/pMH (peptide bound to MH) complexes
Lines 102-103 : the major histocompatibility complex (pMHC). pMHC I and II are indicated. However, for clarity, it is only represented the MHC class I.
> the major histocompatibility (pMH) complex. pMH1 and pMH2 are indicated. However, for clarity, only MH1 is represented.
Lines 106, 114: TCR/CD3ζ > TR/CD247 (CD3 zeta, CD3ζ)
Line 111 : CD3e chain of the TCR > CD3E of the TcR
Lines 120, 125, 127, 132: TCE > bsAb (easier reading)
Line 138: filamentous actin (F-actin) > F-actin
Line 139: TCEs > TCE bsAbs
Lines 144-145: 4-1BB [also known as tumor necrosis factor receptor superfamily member 9 (TNFRSF9) or CD137)] >
TNFRSF9 (tumor necrosis factor receptor superfamily member 9, 4-1BB, CD137)
Line 147: CD40x4-1BB bsAb duobody > CD40xTNFRSF9 bsAb duobody
Line 150: controlx4-1BB > controlxTNFRSF9
Line 151: 4-1BB > TNFRSF9
IMPORTANT : In table please use the HGNC gene name and IMGT/mAb-DB for the targets (for instance CD3E instead of CD3, CEACAM5 instead of CEA, etc) https://www.genenames.org/
Replace GP100 with PMEL peptide 280-288 (presented by HLA*0201)
In the first column, the International Nonproprietary Names (INN) should be in small letters. The four-letter extensions are not part of the INN (it is only relevant for proprietary names approved by the FDA) and can be deleted in this first column.
Line 163: heavy-chain-only immunoglobulins (VHHs)
Line 188, 265 (2x): JK> Jurkat (easier reading)
Line 198: full-lenght > full-length
Line 205: Fc domain, which retains FcRn-driven > Fc, which retains FCGRT (FcRn)-driven
Line 206 : FcγR > FCGR (FcγR)
Line 210: CD8 T cells > CD8+ T cells
L239: CAR > single chain (scFv) fused to … (to be briefly described)
L263 : CAR > CAR-T
Line 264 : showed > led to
